# Effect of Supervised over Self-Performed Eccentric Exercise on Lateral Elbow Tendinopathy: A Pilot Study

**DOI:** 10.3390/jcm11247434

**Published:** 2022-12-15

**Authors:** Antonio Oya-Casero, Miguel Muñoz-Cruzado Barba, Manuel Madera-García, Rosario García-LLorent, Juan Alfonso Andrade-Ortega, Antonio I. Cuesta-Vargas, Cristina Roldán-Jiménez

**Affiliations:** 1Departamento de Fisioterapia, Facultad de Ciencias de la Salud, Universidad de Málaga, Andalucía Tech., Arquitecto Francisco Peñalosa, 3, 29071 Málaga, Spain; 2Unidad de Rehabilitación Puerta de Andalucía, Hospital Universitario de Jaén, 23007 Jaén, Spain; 3Instituto de Investigación Biomédica de Málaga (IBIMA) Plataforma BIONAND, Grupo de Clinimetría (F-14), 29071 Málaga, Spain; 4School of Clinical Science, Faculty of Health Science, Queensland University Technology, Brisbane, QLD 4000, Australia

**Keywords:** exercise therapy, rehabilitation, physical therapy modalities, tendinopathy, elbow tendinopathy, tennis elbow, resistance training

## Abstract

Background: The efficacy of eccentric exercise self-performed by the patient has not been proved in the treatment of lateral elbow tendinopathy (LET). The aim of this study was to compare the effects of a programme of eccentric exercises applied by a physiotherapist to patients with LET through a structured manual programme compared to its self-performance, guided by an illustrated brochure. Method: A single-blind, pilot, randomised, controlled trial was conducted. Twenty patients were randomised. The intervention group carried out eccentric exercises applied directly by a physiotherapist (10 sessions). The control group carried out eccentric exercises that were self-performed by the patient (10 sessions). Both groups received simulated ultrasounds. Pain (visual analogue scale (VAS)), function (DASH questionnaire) and satisfaction (with a Likert scale) were measured at the beginning of the intervention, immediately after the intervention and 3 months after the intervention. Results: The mean age was 53.17 and 54.25 years old. The experimental group presented a greater tendency to improve function (DASH −15.91) and reduce pain (VAS −2.88) compared to the control group, although these differences were not significant (*p* > 0.05). Conclusion: Eccentric exercise, both performed by the physiotherapist and self-performed by the patient, improved function and pain in the patients with LET.

## 1. Introduction

Lateral elbow tendinopathy (LET) is a painful condition included in the so-called enthesopathies [1]. Although this condition is usually included in what is known as “tennis elbow”, practising this sport as a cause of LET only appears in a very low percentage of cases [2]. The prevalence of this disease in the general population is 1–3%, with a peak incidence in people aged between 35 and 59 years [3]. An especially important etiopathogenic factor is the excessive and/or repetitive realisation of activities that involve supination and pronation of the forearm with the elbow extended, causing pain and functional deterioration [4].

It is important to differentiate the term “epicondylitis”, which was used to describe the chronic pain of a symptomatic tendon, involving inflammation as a central pathological process [5], from the term “epicondylosis” LET, which encompasses a structured and degenerative process in which, at the histological level, degenerative phenomena appear, with disorganisation of the collagen fibres [6] and scarce presence of inflammatory cells [7].

The diagnosis of LET is based on the clinical history and physical examination of the patient. Specifically, some authors state that the clinical case must have an evolution of six months, although most researchers establish a cutoff point of three months. The process is not self-limiting, and it is associated with pain and continuous disability in a large number of patients, with “pain when grasping” being the most common functional limitation [8].

The literature has shown that physiotherapy is cost-effective in those interventions that include an active physical approach, such as exercise [9]. For cost-effectiveness in treating LET, Dutch generalist physicians recommend maintaining an expectant attitude toward infiltrations or physiotherapy [10]. However, a recent Australian trial supports manual physiotherapy (8 sessions) over corticoid infiltration, as it found that, although more costly, this was the only effective intervention and, consequently, cost-effective [11]. On the other hand, there is not enough evidence to recommend in LET treatment the use of passive interventions, such as laser therapy, pulsed shortwave diathermy (PSWD), ultrasound, acupuncture and ice, and the treatments with manual and massage therapy and orthopaedic devices have only shown evidence at the beginning of the treatment [12]. These findings drive towards active treatments.

In injury rehabilitation, an active approach helps patients use strategies to overcome limitations and promote patients’ autonomy compared to the passive approach [9]. In physiotherapeutic treatment, eccentric exercise has proved effective in pain reduction and functional improvement in patients who suffer from this disease [13,14,15]. Although evidence is still limited, there is increasing scientific support for using eccentric exercises, especially if these are part of a multimodal treatment programme [16]. A recent meta-analysis shows that eccentric strengthening seems to be most effective for LET [17]. Nowadays, studies have compared eccentric versus concentric exercises self-performed at home [18], eccentric exercise self-performed at home versus no intervention [19] or the additional effect of eccentric exercise self-performed at home in a standard care programme [20]. However, no study has compared the efficacy of eccentric exercises applied by a physiotherapist against exercises self-performed by the patient. The use of one modality or another could have an impact on cost-effectiveness in the treatment of this pathology.

Due to the need for an active approach and the efficacy demonstrated by the eccentric exercise, this pilot study aimed to compare the efficacy of an eccentric exercise programme applied by a physiotherapist to patients with LET through a structured manual programme against the same programme self-performed by the patient.

## 2. Materials and Methods

### 2.1. Participants and Setting

This pilot study included subjects with LET recruited in the University Hospital of Jaen (Spain) through the head physician of the Physical Medicine and Rehabilitation Unit by their primary care physician or specialist. Therefore, all of them belonged to the Andalusian Health Service and they were referred through the normal process of admission to consultation. The inclusion criteria were as follows: aged between 18 and 65 years, with LET of three or more months of evolution. Patients with LET were clinically identified with the following criteria: (1) tenderness over the lateral epicondyle, (2) pain on grasping, (3) pain on passive flexion of the wrist with elbow extension, and (4) pain with resisted wrist extension. The study excluded those patients with generalised musculoskeletal pain, rheumatic conditions involving the upper limb, cervicobrachialgia, traumatic history of the upper limb, neurological pathology or any other pathology that could interfere with the function of the upper limb. The study also excluded patients who were on sick leave or in a litigation situation due to an upper limb pathology.

### 2.2. Ethical Consideration

All subjects agreed to participate voluntarily in the study and gave their written informed consent. The research ethics committee of the province of Jaen (Spain) approved this study following the Declaration of Helsinki. The intervention was registered in Clinical Trials (ID: NCT03996928, OYA-EXC-2019-1).

### 2.3. Sample Size

The recruitment was initiated in January 2017, and it was terminated in March 2020, due to the COVID-19 pandemic. The sample size was calculated a posteriori, using the G Power software. Considering a sample size of *n* = 8 and *n* = 12, with an alpha error of 0.05 and an effect size of 0.44, the statistical power obtained for this pilot study was 0.23.

### 2.4. Outcome Measures

Descriptive and anthropometric variables were measured, such as age (years), sex (male, female), height (m), weight (kg) and BMI (kg/m^2^). To determine the effect of the intervention, the following main outcome variables were measured:

Disability according to the disabilities of the arm, shoulder and hand (DASH) questionnaire. This questionnaire is self-administered and evaluates the upper limb as a functional unit, which allows quantifying and comparing the repercussions of the different processes that affect different regions of such extremity [21]. It is expressed in percent values (0 = no difficulty, 100 = maximum difficulty) [21].

Pain, according to the visual analogue scale (VAS) of 100 mm, validated in LET [22].

Satisfaction, according to a 5-point Likert scale from low to high satisfaction (1 = unsatisfied, 2 = slightly unsatisfied, 3 = neither satisfied nor unsatisfied, 4 = satisfied and 5 = very satisfied) [23,24].

The variables to measure disability and pain were recorded at the beginning of the intervention (baseline), after the intervention (post-intervention) and at 3 months after the completion of the intervention (follow-up). The variable “satisfaction” was measured only immediately and 3 months after the intervention. All assessments were made by the medical doctor.

### 2.5. Procedure

Recruitment started in January 2017 and ended in March 2020. During this period, patients from the Physical Medicine and Rehabilitation Unit who met the criteria were randomly assigned to a group and received the intervention. As the intervention was individualised and did not follow a group format, the intervention in each subject did not coincide with a time milestone.

The independent variables were gathered by the medical doctor, once the patient was recruited for the pilot study and prior to randomisation. The dependent variables were gathered by the same clinician at three different time points: (1) the day that the study was initiated (blinded with respect to the group that each patient was randomised to); (2) 2 weeks after the completion of the treatment, coinciding with the day of the last session of physiotherapy; and (3) at 3 months after the completion of the treatment, at an arranged visit in external office (see Figure 1). All data were loaded into an Excel sheet for subsequent analysis by the medical doctor.

The subjects were randomised into two possible groups: intervention group and control group. To this end, we used an open-access software (Nosetup.org © 2022-2022, http://nosetup.org/php_on_line/numero_aleatorio, last access on 17 November 2022), which allows obtaining a random sequence of two possible numbers (1 and 2), with 1 corresponding to the intervention group and 2 referring to the control group. No group stratifications were performed. Patients were initially assessed by the medical doctor to ensure inclusion criteria. Post-intervention and follow-up assessment were also made by the medical doctor, who was blinded to group assignment. Only the physiotherapy supervisor knew the sequence and was in charge of randomising the patients to each group, and she did not participate in the treatment or analysis of the data.

Two physiotherapists applied the sessions: one in the control group and one in the intervention group. They were not informed about which group was the intervention or control group, and none of them participated in the gathering or analysis of the data. Therefore, participants and physiotherapists were blinded to group assignment to reduce risk of vias. In the control group, the physiotherapist carried out a supervision to improve the adherence to the treatment through the application of simulated ultrasounds, as well as to guarantee the realisation of the exercise programme.

### 2.6. Interventions

#### 2.6.1. Experimental Group

The intervention consisted of mobilisation exercises, stretching exercises and eccentric exercises. Both the mobilisation and stretching exercises are detailed in Annex I. The epicondylar eccentric exercises were applied directly by a physiotherapist as follows: (i) starting position (Figure 2A): the patient leans his/her forearm in elbow pronation and flexion at 100° on the treatment table, leaving the hand outside, and in neutral wrist flexo-extension; (ii) activity (Figure 2B): firstly, the physiotherapist passively moves the wrist-hand of the patient to dorsal flexion with radial tilt and, from this position, the physiotherapist exerts an approximate pressure of 2 kg on the back of the hand, producing the palmar flexion with ulnar tilt, requesting the patient to oppose to it, so that the final position is reached very slowly and in a controlled manner in 10 s; (iii) once the final position is reached (Figure 2C), the physiotherapist moves the hand of the patient to the starting position, in order to avoid the concentric activity. The sequence that was carried out consisted of 3 sets of 10 repetitions each [9,25]. A total of 10 sessions were carried out, which lasted 20 min each, distributed in one session per day from Monday to Friday for a total period of 2 weeks; moreover, simulated ultrasound therapy was administered to the patients in all ten sessions, with the aim of improving their adherence.

#### 2.6.2. Control Group

A physiotherapist briefed the patients of the control group on an exercise plan equivalent to that performed in the intervention group, with the aid of an illustrated brochure, which was given to the patient for guidance. Since the patients in this group were not assisted by a physiotherapist, the palmar flexion was self-performed with the aid of an elastic band, in the following manner: (i) the patient, sitting on a chair, leans the forearm in elbow pronation and flexion at 100°, leaving the hand outside, and in neutral wrist flexo-extension; (ii) with the elastic band in tension, the patient moves the hand slowly and progressively toward palmar flexion with cubital tilt; then, (iii) the patient returns to the dorsal flexion position and radial tilt, using the elastic band to avoid a concentric exercise (Figure 3). The mobilisation and stretching exercises were the same as those carried out in the experimental group (Table A1). In addition to the self-performed exercise, simulated ultrasound therapy was administered to the patients in all ten sessions to improve their adherence to the treatment. Attendance to the simulated ultrasound also enabled the follow-up of the patient for the completion of the home treatment.

### 2.7. Statistical Analysis

Descriptive statistics were calculated for all outcome measures, including measurements of central tendency and dispersion. Qualitative outcomes were described as number of subjects (*n*) and percentage (%). Normal distribution was tested using the Kolmogorov–Smirnov test. The effect was established for clinical outcomes by comparing scores at the end of the intervention and at a 3-month follow-up. Depending on the result, either a Student’s *t*-test or a Wilcoxon’s test was applied. Within-group changes were also examined and described by mean and standard error (SE). Analyses were performed using SPSS version 22.0 (SPSS Inc., Chicago, IL, USA).

## 3. Results

A total of 25 patients were initially recruited, and finally, 20 of them were included and randomised. More details are given in Figure 4.

Table 1 shows the sample baseline outcomes. No significant differences were found between the groups in clinical outcome measures at baseline. All variables presented a normal distribution (*p* > 0.05).

Table 2 shows the results after the intervention, the intra-group changes and the differences in means between groups. Both groups showed a decrease in the values of DASH and VAS, with these changes being greater in the experimental group. In the experimental group, DASH decreased by 15.91 points, and VAS decreased by 2.88 on average, whereas in the control group, DASH and VAS decreased by 8.7 and 1.4 points, respectively. However, these differences were not significant. More details are shown in Table 2.

Table 3 shows the within-group changes and differences between groups in DASH and VAS in the 3-month follow-up. Although both groups experienced a decrease in the DASH and VAS scores at 3 months, these changes were greater in the control group, although without significant differences between groups. Further details are shown in Table 3.

The results of the variable satisfaction are shown in Table 4. In both groups, the patients showed satisfaction with the treatment received.

## 4. Discussion

The main finding of this pilot study was that the scheduled eccentric exercise for LET was effective in the treatment of this pathology, as it improved the pain and functionality of the elbow in the patient. The experimental group presented a greater tendency to functional improvement and pain reduction with respect to the group of patients who self-performed the same exercise programme, although these differences were not significant, possibly due to the sample size. Moreover, in the follow-up, this tendency changed, with the control group showing greater functionality; however, pain reduction was still greater in the experimental group.

The eccentric exercise performed on the patient manually by the physiotherapist produced an improvement in the functionality of the upper limb of the patient, as is shown by the decrease of 15.91 and 2.88 points in the DASH and VAS scales, respectively, after the intervention (Table 2). In the follow-up, this difference increased by 4.26 and 0.6 points in DASH and VAS, respectively (Table 3).

The eccentric exercise has been previously shown as an effective alternative in the management of LET [26]. In general, it is agreed that strength training decreases the symptoms of tendinosis [27], which corroborates the effect found in our study.

The improvement in elbow functionality and pain in both groups is in line with that found in the study of Anitha et al. [28]. In that study, 60 subjects were randomised into two groups of 30 subjects each. The control group only received ultrasound and the experimental group received ultrasound and the eccentric exercise of the epicondylar musculature. The results were favourable for the experimental group, as it showed a statistically significant improvement in terms of pain from 5.5 to 4.13 points (*p* < 0.001), measured with the numerical pain scale, and an improvement in grip strength, measured by dynamometry from a mean value of 15.60 kg to 24.56 kg (*p* < 0.001).

The results of the present study are also in agreement with those of the pilot study of Tyler et al. [20], in which 21 subjects were randomised into two groups: one with 11 subjects, who performed eccentric training, and the other with 10 subjects, who received a standard physiotherapy treatment. The programme of eccentric exercises proved to be effective in the treatment of LET, obtaining that all dependent variables were greater for the eccentric group compared to the group with the standard treatment in DASH (76% for the eccentric training group vs. 13% for the control group) and VAS (81% for the eccentric training group vs. 22% for the control group). In the present study, both groups performed eccentric work, obtaining improvements in DASH and VAS in both groups (Table 2 and Table 3).

The home eccentric work through an exercise table taught to the patient by a physiotherapist also proved to be effective in terms of pain and functionality. Specifically, it also improved the functionality of the upper limb, as it reduced DASH and VAS by 8.7 and 1.4 points, respectively, after the intervention (Table 2). In the follow-up, this difference increased by 7.97 and 0.13 points in DASH and VAS, respectively (Table 3). Therefore, the finding of this study is in line with that of studies such as that of Söderberg et al. [19]. In the mentioned study, 42 subjects were randomised into two groups: the experimental group performed a home eccentric training with an epicondylar band, whereas the control group only received an epicondylar band, with a 6-week intervention in both groups. It was concluded that a daily intervention of eccentric exercises performed at home was effective to increase hand strength without pain, reducing the cases of lateral epicondylalgia during the intervention. After the intervention, with respect to the control group, the subjects of the experimental group showed a significantly greater hand grip strength (F = 5.51, *p* = 0.025) and hand extension strength (F = 10.39, *p* = 0.0001) without pain.

The effect of the home eccentric exercise for LET has been previously addressed in more studies. Nilsson et al. [29] conducted a study of 78 patients with LET in an intervention group with 51 patients, applying home training, ergonomic advice and bandages whenever necessary; the control group received the standardised treatment (anti-inflammatory drugs, corticosteroids, physiotherapy, bandages, etc.). After four weeks, the intervention group showed fewer sick leaves for LET, reducing pain and improving functionality, which were measured with the forearm evaluation questionnaire PRFEQ; moreover, they returned to their jobs earlier than the subjects in the control group.

Furthermore, the effect of eccentric exercises has been studied in other tendinopathies, such as the Achilles tendon. Arnal-Gómez et al. [30] carried out a literature review in which they observed that the eccentric exercise therapy applied to Achilles tendinopathy was effective by itself and in combination with other treatments; this suggests that the intervention programme based on eccentric exercise is the best way to treat Achilles tendinopathy. Stevens et al. [31] used two different ways of conducting eccentric exercise in 28 participants; Alfredson’s protocol of eccentric exercises was taught to both groups, although one of the groups was requested to carry out 180 repetitions, and the other group was asked to perform as many repetitions as they could. Both groups improved their scores in VISA-A and VAS after 6 weeks. Specifically, the VAS scale in the experimental group decreased by 2.39 mm after the intervention, which was less than in the experimental group of the present study. Lastly, Yu et al. [32], in their study with 32 subjects randomised into two groups, compared the eccentric exercise (experimental group) and the concentric exercise (control group). As a result, the experimental group showed a significant improvement in pain of 3.56 points, measured with the VAS scale, which is similar to the results found in our study, in which we obtained a decrease of 2.88 points in VAS (Table 2).

These mentioned studies included eccentric exercise and other therapies as part of the treatment. Similarly, the present study also included stretching, simulated ultrasound, and exercise progression, following the usual treatment in the Rehabilitation Unit. Although it allows studying possible differences due solely to the modality of application of eccentric exercises, comparing between studies in the literature is not easy. Regarding satisfaction, 75% of the patients stated that they were satisfied with the treatment after the intervention (Table 4). Braaskma et al. [23] also measured satisfaction. In their study, 45.4% claimed to be satisfied with the treatment they received in terms of pain reduction and 56.4% in the improvement of functionality.

We found differences both in the interventions and in the design in all the mentioned studies. However, in the present study, the role of the physiotherapist was especially relevant in the management of these patients, since he/she was in charge of teaching the technique to the patient in one way or the other, and the physiotherapist’s institution was in charge of deciding how to apply such exercises depending on the patients’ needs.

Subjects from previous research may be similar to the present study’s sample, as subjects were included if they had a minimum period of symptoms and positive clinical tests. For example, Tyler et al. included subjects with symptoms for more than six weeks. They had been diagnosed using several tests, such as pain on palpation at the lateral epicondyle, among other clinical tests. [20]. Along the same line, subjects from other studies presented symptoms for at least one month and pain during several clinical tests [19]. In addition, their mean age ranged between 47 and 51 years old, similar to the preset sample (mean age 54 and 53 years old).

The present study has some limitations that must be pointed out. Firstly, since it is a pilot study, it is necessary to carry out randomised clinical trials that include a larger number of subjects. Secondly, it would be interesting to study the effects of each type of intervention separately, including a group that receives the eccentric exercise manually by a physiotherapist, another group that self-performs the eccentric exercise following an illustrated brochure, and a third placebo control group. Thirdly, given that the follow-up period from the start of the study is 12 weeks and that LET has a high recurrence rate, the current results should be seen as evidence of the short-term efficacy of eccentric strengthening. Therefore, whether the treatment approach provides similar long-term efficacy remains to be determined. Finally, although the execution of the eccentric exercise is easy to apply manually, there could be differences in the application of strength by the physiotherapist. Lastly, as this is a pilot study, results should be taken with caution and may not be generalisable to the population suffering from LET. Future studies should address these limitations.

## 5. Conclusions

A programme of 10 sessions of eccentric exercises, conducted by a physiotherapist or self-performed by the patient after learning the technique, produced improvements in upper limb function and pain in patients with LET. Although there was a tendency for improvement in the experimental group, who performed the exercises under the supervision of the physiotherapist, there were no significant improvements between the two groups. Future clinical trials with larger samples are necessary to determine the existence of differences between these two interventions.

## Figures and Tables

**Figure 1 jcm-11-07434-f001:**
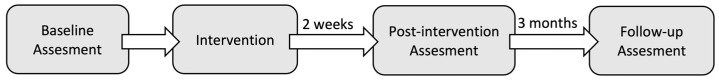
Dependent outcomes were assessed at three different times: Baseline, post-intervention and follow-up.

**Figure 2 jcm-11-07434-f002:**
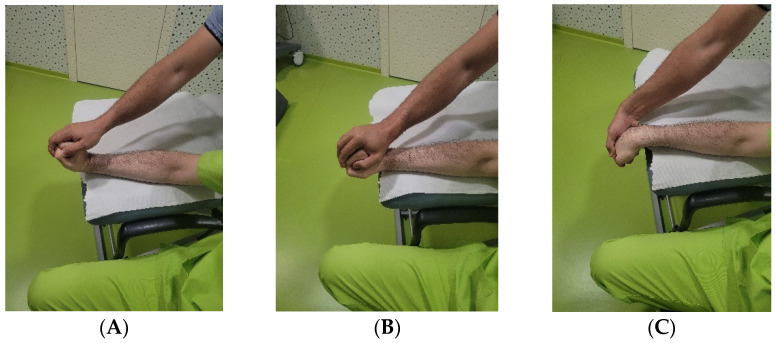
Patient performing the exercises in the intervention group. (**A**): Starting position. (**B**): Intermediate position. (**C**): Final position.

**Figure 3 jcm-11-07434-f003:**
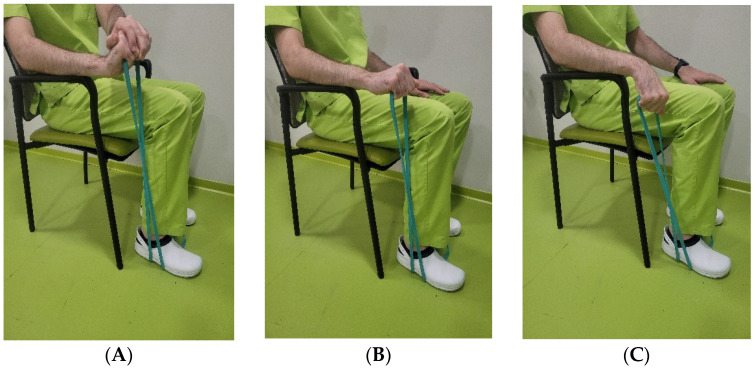
Patient in the control group performing the exercise using the elastic band: (**A**) starting position; (**B**) intermediate position; (**C**) final position.

**Figure 4 jcm-11-07434-f004:**
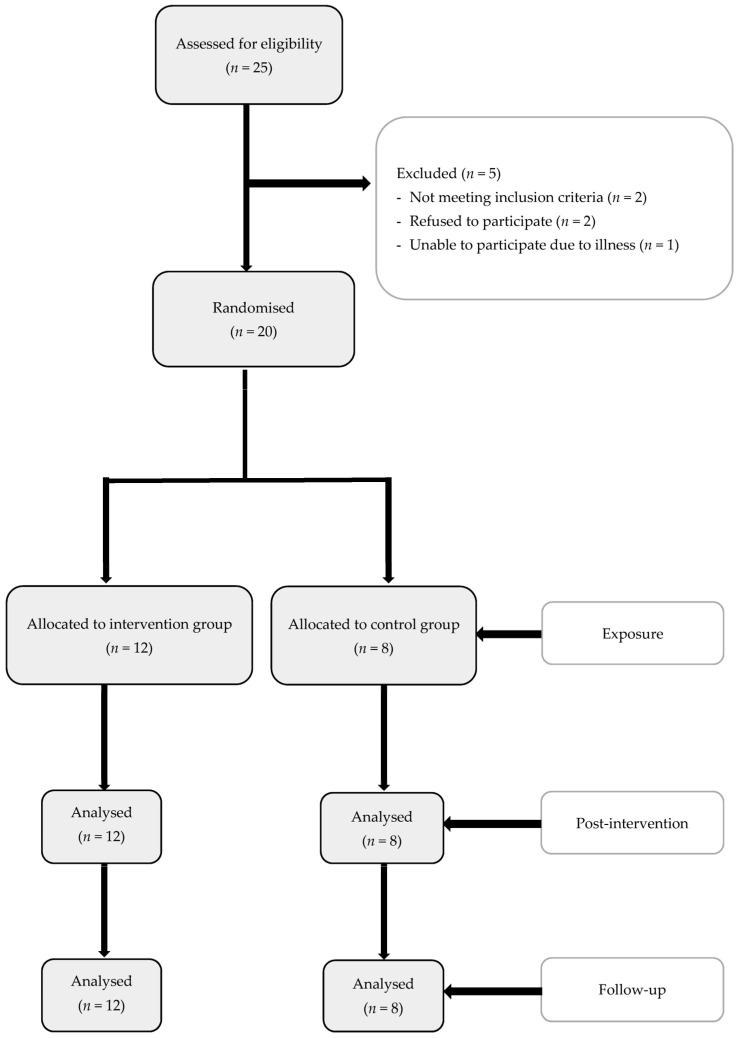
Diagram for the pilot study.

**Table 1 jcm-11-07434-t001:** Comparison between groups at baseline.

	Experimental Group (*n* = 8)	Control Group (*n* = 12)	*p* Values
Age (years)	54.25 (5.53)	53.17 (10.72)	0.771
Height (m)	1.65 (1.80–1.49)	1.63 (1.73–1.51)	0.678
Weight (kg)	76 (9)	76.36 (19.98)	0.962
BMI (kg/m^2^)	23.64 (0.11)	23.27 (5.30)	0.832
DASH	54.25 (18.36)	56.42 (15.70)	0.781
VAS	6.5 (2.35)	6.04 (1.73)	0.557

DASH: disabilities of the arm, shoulder and hand (disability); VAS: visual analogue scale (pain).

**Table 2 jcm-11-07434-t002:** Post-intervention results of primary outcome measures including within-group and between-group changes.

	Post-Intervention		Within-Group Changes		Between-Group Changes
	Intervention Group (*n* = 8)	Control Group (*n* = 12)	Intervention Group (*n* = 8)	Control Group (*n* = 12)	
	Mean (SD)	Mean (SD)	Mean Difference (SD)	Mean Difference (SD)	Mean (SE)
DASH	38.34 (20.41)	47.72 (21.99)	−15.91 (15.17)	−8.7 (20.27)	7.20 (7.93)
VAS	3.7 (2.36)	4.62 (2.75)	−2.88 (1.87)	−1.4 (2)	1.47 (0.88)

DASH: disabilities of the arm, shoulder and hand (disability); VAS: visual analogue scale (pain).

**Table 3 jcm-11-07434-t003:** Follow-up results of primary outcome measures including within-group and between-group changes.

	Follow-Up (3 Months)		Within-Group Changes		Between-Group Changes
	Intervention Group (*n* = 8)	Control Group (*n* = 12)	Intervention Group (*n* = 8)	Control Group (*n* = 12)	
	Mean (SD)	Mean (SD)	Mean (SD)	Mean (SD)	Mean (SE)
DASH	34.08 (22.59)	31.05 (16.25)	−4.26 (14.70)	−16.66 (10.76)	−12.40 (6.05)
VAS	3.1 (2.10)	3.33 (1.96)	−0.51 (1.15)	−1.29 (1.32)	−0.779 (0.55)

DASH: disabilities of the arm, shoulder and hand (disability); VAS: visual analogue scale (pain).

**Table 4 jcm-11-07434-t004:** Description of satisfaction after the intervention and at the 3-month follow-up in both groups.

Satisfaction	Post-Intervention		Follow-Up (3 Months)	
	Intervention Group (*n* = 8)	Control Group (*n* = 12)	Intervention Group (*n* = 8)	Control Group (*n* = 12)
Unsatisfied	1 (12.5%)	1 (8.3%)	1 (12.5%)	-
Slightly unsatisfied	-	-	7 (87.5%)	-
Neither satisfied nor unsatisfied	-	-	-	
Satisfied	6 (75%)	9 (75%)	-	6 (50%)
Very satisfied	1 (12.5%)	2 (16.7%)	-	6 (50%)

## Data Availability

Data are available on request from the corresponding author.

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
