# Peer review of "Effect of Supervised over Self-Performed Eccentric Exercise on Lateral Elbow Tendinopathy: A Pilot Study"

_jcm, 2022, doi:10.3390/jcm11247434_

Round 1

Reviewer 1 Report

Dear Author,
I value the opportunity to review the current manuscript, and I hope my comments will improve the quality of the article. My comments are attached as a word document. 

Thanks & regards

Reviewer

Author Response

Manuscript Review Report

Submitted to: Sports Medicine

Manuscript ID: jcm-2074574

Title: Effect of eccentric exercise performed by a physiotherapist compared
to its self-realisation by the patient: A pilot study,

Title: 

Novelty & originality: Fair

Topic not reflect the content: P1-L1: Title: current title is not clear rather confusing, rewrite it as "efficacy of supervised over self-performed eccentric exercise on epicondylosis'

Authors: Thank you. This comment is closely related to comments made by reviewer 2 and 3. The term epicondylosis in the manuscript was replaced by lateral elbow tendinopathy (LET). And, and proposed by also reviewer 2, it was added to the title. Aditionaly, the term self-realisation was replaced by self-performed for better consistency in the manuscript. After changes made, and taken into account your recommendation, the new title is presented as follows:

“Effect of supervised over self-performed Eccentric Exercise on Lateral Elbow Tendinopathy: A Pilot Study”

We do really appreciate this comment, because the title now is easier to follow.

Abstract:

Clinical significance: P1 L36: missing

Authors: Thank you. This comment concurs with reviewer 1, who asked us for more details in the abstract. However, if we add new information, we must remove some key information given the word limitation.

Introduction:

P1-L3: the use of term 'epicondylosis' needs more clarity on inclusion criteria based on definition (P2 L98) histological level changes?!

Authors: Thank you. In concordance with reviewer 2, we have added more details to inclusion criteria. Now, that paragraph appears as follows:

(…) “The inclusion criteria were as follows: aged between 18 and 65 years, with LET of three or more months of evolution. Patients with LET were clinically identified with the following criteria: 1) tenderness over the lateral epicondyle, 2) pain on grasping, 3) pain on passive flexion of the wrist with elbow extension, and 4) pain with resisted wrist extension”

Methods:

Consent and IRB: stated

Did you select study subjects based on histological changes criterion? OR how could you diagnose as epicondylosis?

Authors: Thank you. The information for this comment is provided in the previous one (more details about inclusion criteria).

P2 L98: inclusion criteria, need more clarity, stages of epicondylosis acute/subacute/chronic? What is the occupational background of selected subjects, are they sports persons/skilled workers?

Authors: Thank you. This comment is closely related to the previous one. Regarding the occupational background, authors do not have this information. However, as we are willing to do a randomized controlled trial with a wider sample, we take note and this data will be included in future research.

P4 L162: approx. pressure of 2 kg?! How could you justify the equalization/standardization of same pressure on each subject and between groups in repetitions.

Authors: Thank you. We do agree with you, as it is a limitation of the study. Regarding inter-subject variability, it is a limitation of the intervention itself. Regarding intra-subject variability, the physiotherapist was the same person in the intervention group. This limitation has already been contemplated in the manuscript, as follows:

“Finally, although the execution of the eccentric exercise is easy to apply manually, there could be differences in the application of strength by the physiotherapist.”

What was the US dose?

Authors: Thank you. None. In both groups, US was simulated to facilitate adherence.

simulated ultrasound therapy was administered to the patients in all ten session, with the aim of improving their adherence.”.

Data & result analysis: 

Appropriate tests used & well explained Discussion: 

P7 L248: 'our study’?! was it a mistake? Because 30 subjects in each group?! need to cross check, it is different from abstract (20 subjects) L249 control group protocol?!

Authors: Thank you so much for this comment. It is a typing mistake. It should be “In that study” instead of “our”. We have already modified it.

list some demographic/subject character specific differences between previous studies and current results

Authors: Thank you. We added the following paragraph in the discussion:

“Subjects from previous research may be similar to the present study's sample, as subjects were included if they had a minimum period of symptoms and positive clinical tests. For example, Tyler et al. included subjects with symptoms for more than six weeks. They had been diagnosed using several tests, such as pain on palpation at the lateral epicondyle, among other clinical tests. [19]. Along the same line, subjects from other studies presented symptoms for at least one month and pain during several clinical tests [18]. In addition, their mean age ranged between 47 and 51 years old, similar to the preset sample (mean age 54 and 53 years old).”

P7 L278: Need to justify the reasoning for progress on both groups and role of other PT treatments such as US/stretching/exercises on study variables.

Authors: Thank you. The following information was added:

“These mentioned studies included eccentric exercise and other therapies as part of the treatment. Similarly, the present study also included stretching, simulated ultrasound, and exercise progression, following the usual treatment in the Rehabilitation Unit. Although it allows studying possible differences due solely to the modality of application of eccentric exercises, comparing between studies in the literature is not easy.”

Conclusion: 

Clinical implication need to be explain clearly

Authors: Thank you. We cannot affirm a clear clinical implication because it is a pilot study. In the conclusion, we mention that “Although there was a tendency to improvement in the experimental group, who performed the exercises under the supervision of the physiotherapist, there were no significant improvements between the two groups”. In the manuscript, with detailed that this finding may be due to the sample size. Therefore, we concluded that “Future clinical trials with larger samples are necessary to determine the existence of differences between these two interventions” in the last line.

COI/Fund statement: Yes

References: Latest articles cited

Tables: satisfactory

Reviewer 2 Report

Reviewer Comments

Thank you very much for the opportunity to review the manuscript submission entitled: “Effect of Eccentric Exercise Performed by a Physiotherapist Compared to Its Self-Realisation by the Patient: A Pilot Study.

The aim of this study was to compare the effects of a programme of eccentric exercises applied by a physiotherapist to patients with epicondylosis through a structured manual programme compared to its self-realisation, guided by an illustrated brochure. The study is interesting; however, some limitations and constructive comments are pointed out below:

Title: Good

Abstract:

·      Include the mean age of the participants.

·      Include the condition of the participants in the title.

·      Include MeSH terms as keywords

·      Include p values for statistical differences for within-group differences and between-group differences

·      What is the difference between self-realization and self-performed? Be clear in using the words; it is confusing.

Introduction

·      Explain the scientific background and rationale for the investigation being reported. Needs more justification for comparing these two approaches, i.e. eccentric exercise individualized or supervised.

·      Include prespecified hypotheses

Methods

·      Present key elements of study design early in the paper

·      Describe the setting, locations, and relevant dates, including periods of recruitment, exposure, follow-up, and data collection

·      The inclusion and exclusion criteria should be emphasized.

·      The randomization process should be described in detail.

·      Clearly define all outcomes, potential confounders, and effect modifiers

·      Describe any efforts to address potential sources of bias

Results and discussion

·      Participant flow should be provided using a flowchart - For each group, the numbers of participants who were randomly assigned received intended treatment, and were analysed for the primary outcome

·      Describe any methods used to examine subgroups and interactions

·      For each primary and secondary outcome results for each group and the estimated effect size and its precision (such as a 95% confidence interval) should be reported.

·      Trial limitations, addressing sources of potential bias, imprecision, and the multiplicity of analyses should be discussed

·      Generalisability (external validity, applicability) of the trial findings should be discussed

·      Interpretation consistent with results, balancing benefits and harms, and considering other relevant evidence should be considered.

Author Response

REVIEWER #2

Reviewer Comments

Thank you very much for the opportunity to review the manuscript submission entitled: “Effect of Eccentric Exercise Performed by a Physiotherapist Compared to Its Self-Realisation by the Patient: A Pilot Study.

The aim of this study was to compare the effects of a programme of eccentric exercises applied by a physiotherapist to patients with epicondylosis through a structured manual programme compared to its self-realisation, guided by an illustrated brochure. The study is interesting; however, some limitations and constructive comments are pointed out below:

Title: Good

Abstract:

  • Include the mean age of the participants.

Authors: Thank you so much. This information is not provided in the abstract due to words limitation.

  • Include the condition of the participants in the title.

Authors: Thank you so much for this comment. We have also introduced recommendations by reviewer 1, and the title now appears as follows:

Effect of Supervised over Self-Performed Eccentric Exercise on Lateral Elbow Tendinopathy: A Pilot Study

In addition, the term epicondilosys was replaced by lateral elbow tendinopathy (LET) along the whole manuscript, in accordance with reviewer 3.

  • Include MeSH terms as keywords

Authors: Thank you so much. We have revised the keywords and some of them have been replaced by MeSH terms. 

  • Include p values for statistical differences for within-group differences and between-group differences

Authors: Thank you. As any finding was statistically significant, we did not reported specific p values higher than 0.05.

  • What is the difference between self-realization and self-performed? Be clear in using the words; it is confusing.

Authors: Thank you so much for this comment. We have replaced self-realization by self-performed in the manuscript.

Introduction

  • Explain the scientific background and rationale for the investigation being reported. Needs more justification for comparing these two approaches, i.e. eccentric exercise individualized or supervised.

Authors: Thank you so much for this comment. Authors have reviewed the introduction section. We have modified the text ordering some information, removing some paragraph and adding new information. The justification of the present study can be read as follows:

“The literature has shown that physiotherapy is cost-effective in those interventions which include an active physical approach, such as exercise [9]. For cost-effectiveness in treating LET, Dutch generalist physicians recommends to maintaining an expectant attitude toward infiltrations or physiotherapy [10]. However, a recent Australian trial supports manual physiotherapy (8 sessions) over corticoid infiltration, as it found that, although more costly, this was the only effective intervention and, consequently, cost-effective [11]. On the other hand, there is not enough evidence to recommend in LET treatment the use of passive interventions, such as laser therapy, pulsed shortwave diathermy (PSWD), ultrasound, acupuncture, and ice, and the treatments with manual and massage therapy and orthopedic devices have only shown evidence at the beginning of the treatment [12]. These findings drive towards active treatments.

In injury rehabilitation, an active approach helps patients use strategies to overcome limitations and promote patients’ autonomy compared to the passive approach [9]. In physiotherapeutic treatment, eccentric exercise has proved effective in pain reduction and functional improvement in patients who suffer from this disease [13–15]. Although evidence is still limited, there is increasing scientific support for using eccentric exercises, especially if these are part of a multimodal treatment programme [20]. A recent meta-analysis shows that eccentric strengthening seems to be most effective for LET [16]. Nowadays, studies have compared eccentric versus concentric exercises self-performed at home [17], eccentric exercise self-performed at home versus no intervention [18] or the additional effect of eccentric exercise self-performed at home in a standard care programme  . However, no study has compared the efficacy of eccentric exercises applied by a physiotherapist against self-performed by the patient. The use of one modality or another could have an impact on cost-effectiveness in the treatment of this pathology”

New reference:

  1. Chen, Z.; Baker, N.A. Effectiveness of Eccentric Strengthening in the Treatment of Lateral Elbow Tendinopathy: A Systematic Review with Meta-Analysis. J. Hand Ther. 2021, 34, 18–28, doi:10.1016/j.jht.2020.02.002.

. Include prespecified hypotheses

Authors: Thank you. The present manuscript is a pilot study. Therefore, according to CONSORT guidelines for pilot study, a hypotheses is not recommended (“Formal hypothesis testing for effectiveness (or efficacy) is not recommended”), and authors can only report specific objectives or research question and estimate a treatment effect, as done in the present study.

Reference: Eldridge SM, Chan CL, Campbell MJ, Bond CM, Hopewell S, Thabane L, Lancaster GA. and on behalf of the PAFS consensus group. CONSORT 2010 statement: extension to randomised pilot and feasibility trials. Pilot Feasibility Study. 2016;2:64. https://doi.org/10.1186/s40814-016-0105-8.

Methods

  • Present key elements of study design early in the paper

Authors: Thank you. We have included the design of the present study (pilot study) in the first line of Methods sections, right after “Participants and setting” subheading.

  • Describe the setting, locations, and relevant dates, including periods of recruitment, exposure, follow-up, and data collection

Authors: Thank you for this comment, as we did notice the procedure was not detailed enough.

      New information (including two new figures) has been provided in methods section:

“Recruitment started in January 2017 and ended in march 2020. During this period, patients from the Physical Medicine and Rehabilitation Unit who met the criteria were randomly assigned to a group and received the intervention. As the intervention was individualized and it did not follow a group format, the intervention in each subject did not coincide with a time milestone (see figure 1).

Figure 1. Period of recruitment and intervention.

The dependent variables were gathered by the same clinician at three different time points: 1) the day that the study was initiated (blinded with respect to the group that each patient was randomised to); 2) 2 weeks after the completion of the treatment, coinciding with the day of the last session of physiotherapy; and 3) at 3 months after the completion of the treatment, at an arranged visit in external office (see figure 2). All data were loaded into an Excel sheet for subsequent analysis.

Figure 2. Dependent outcomes were assessed at three different times: Baseline, post-intervention and follow-up. 

  • The inclusion and exclusion criteria should be emphasized.

Authors: Thank you. Inclusion and exclusion criteria have been detailed with new information:

“…by their primary care physician or specialist. Therefore, all of them belonged to the Andalusian Health Service and they were referred through the normal process of admission to consultation”

“Patients with epicondylosis were clinically identified with the following criteria: 1) tenderness over the lateral epicondyle, 2) pain on grasping, 3) pain on passive flexion of the wrist with elbow extension, and 4) pain with resisted wrist extension”

  • The randomization process should be described in detail.

In the manuscript, authors detailed how participants were allocated to interventions and have added new information about masking, based on CONSORT guidelines. The whole paragraph now can be read as follows:

The subjects were randomised to two possible groups: intervention group and control group. To this end, we used an open-access software (http://nosetup.org/php_on_line/numero_aleatorio), which allows obtaining a random sequence of two possible numbers (1 and 2), with 1 corresponding to the intervention group and 2 referring to the control group. Patients were initially assessed by the medical doctor to ensure inclusion criteria. Post-intervention and follow-up assessment were also made by the medical doctor, who was blinded to group assignment. Only the physiotherapy supervisor knew the sequence and was in charge of randomising the patients to each group, and she did not participate in the treatment or analysis of the data.

Two physiotherapists applied the sessions: one in the control group and one in the intervention group. They were not informed about which group was the intervention or control group, and none of them participated in the gathering or analysis of the data. Therefore, participants and physiotherapists were blinded to group assignment. In the control group, the physiotherapist carried out a supervision to improve the adherence to the treatment through the application of simulated ultrasounds, as well as to guarantee the realisation of the exercise programme.

  • Clearly define all outcomes, potential confounders, and effect modifiers

Authors: Thank you. Outcomes are defined in the methods section (outcome subheading) and potential cofounders are provided in limitations of the study. Limitations from this pilot study will be taken into account in future randomized controlled trials.

  • Describe any efforts to address potential sources of bias

Authors: Thank you. To reduce risk of vias, both patients, the medical doctor who did the recruitment and the patients were blinded. This information has been detailed in the manuscript.

Results and discussion

  • Participant flow should be provided using a flowchart - For each group, the numbers of participants who were randomly assigned received intended treatment, and were analysed for the primary outcome

Authors: Thank you so much for this comment. We have provided a flowchart in results section (figure 5). As you can see, there was any lost, so all the patient who joined the study ended the intervention in both groups.

A total of 25 patients were initially recruited, and finally 20 of them were included and randomized. More details are given in figure 5

(figure 5 bout here. Please see in the document)

        Figure5. Diagram for the pilot study.

  • Describe any methods used to examine subgroups and interactions

            Authors: Thank you. This is described in Statistical Analysis subheading.

  • For each primary and secondary outcome results for each group and the estimated effect size and its precision (such as a 95% confidence interval) should be reported.

            Authors: Thank you. Effect size is not applicable in pilot studies.

Reference: Reference: Eldridge SM, Chan CL, Campbell MJ, Bond CM, Hopewell S, Thabane L, Lancaster GA. and on behalf of the PAFS consensus group. CONSORT 2010 statement: extension to randomised pilot and feasibility trials. Pilot Feasibility Study. 2016;2:64. https://doi.org/10.1186/s40814-016-0105-8.

  • Trial limitations, addressing sources of potential bias, imprecision, and the multiplicity of analyses should be discussed
  • Generalisability (external validity, applicability) of the trial findings should be discussed
  • Interpretation consistent with results, balancing benefits and harms, and considering other relevant evidence should be considered.

            Authors: Thank you. Information regarding these three comments is provided below:

The present study has some limitations that must be pointed out. Firstly, since it is a pilot study, it is necessary to carry out randomised clinical trials that include a larger number of subjects. Secondly, it would be interesting to study the effects of each type of intervention separately, including a group that receives the eccentric exercise manually by a physiotherapist, another group that self-performs the eccentric exercise following an illustrated brochure, and a third placebo control group. Thirdly, given that the follow-up period from the start of the study is 12 weeks and that LET has a high recurrence rate, the current results should be seen as evidence of the short-term efficacy of eccentric strengthening. Therefore, whether the treatment approach provide similar long-term efficacy remains to be determined. Finally, although the execution of the eccentric exercise is easy to apply manually, there could be differences in the application of strength by the physiotherapist. Lastly, as this is a pilot study, results should be taken with caution and may not be generalizable to the population suffering from LET. Future studies should address these limitations

Reviewer 3 Report

The term epicondylosis is not correct.

The physical agents/electrotherapy modalities are ineffective as sole treatments. 

You do not use the appropriate outcome measures such asPRTEE AND GRIP STRENGTH.

The eccentric training belongs to the past. It is used an exercise program for the whole arm. Expain why

Author Response

REVIEWER #3  RESPONSE.

The term epicondylosis is not correct.

Authors: Thank you. We do agree with you. Although the term epicondylosis or lateral epicondylosis is widely used ((Raman et al., 2012), (Li et al., 2022), (Bureau et al., 2022), (Stegink-Jansen et al., 2021)). The term lateral elbow tendinopathy (LET) (Chen & Baker, 2021) has been used in the whole manuscript.   

The physical agents/electrotherapy modalities are ineffective as sole treatment,

Authors: Thank you. We do agree with you, as you can read in the introduction:

On the other hand, there is not enough evidence to recommend in LET treatment the use of passive interventions, such as laser therapy, pulsed shortwave diathermy (PSWD), ultrasound, acupuncture, and ice, and the treatments with manual and massage therapy and orthopedic devices have only shown evidence at the beginning of the treatment [12]

As evidence does not support the use of these techniques for the treatment of LET, we studied the effect of an active approach such as eccentric exercises on LET.

You do not use the appropriate outcome measures such as PRTEE AND GRIP STRENGTH.

Authors: Thank you. In this study, we used DASH questionnaire and VAS, which are appropriate for measuring upper limb function and pain, respectively. Those patient-reported outcomes have been validated and previously used in this population. In fact, a recent systematic review (Chen & Baker, 2021) report that the scales more used to measure functionality in studies were: : DASH, TEFS, PRTEE, PRFEQ y EVA. However, it would be interesting to include those measure in future studies aimed at measuring changes in grip strength.

The eccentric training belongs to the past. It is used an exercise program for the whole arm. Expain why

Authors: Thank you. The rationale of the intervention is provided in the introduction section. Please find more information detailed below:

Eccentric exercise can be a key component of tendinopathy rehabilitation if integrated at the appropriate stage of injury. In the rehabilitation of tendinopathies, eccentric exercise not only provides structural and functional benefits, but also neuromuscular benefits through central adaptation of agonist and antagonist muscles (Page, 2010). Studies have shown that isolated eccentric strength training is effective in treating Achilles, patella, and shoulder tendinopathy (Purdam, 2004; Williamson & Hoggart, 2005). In patients with LET, eccentric exercise reduces pain and increases strength in LET more effectively than concentric strengthening exercise (Peterson et al., 2014).

These information is also supported by more recent systematic reviews: The systematic review from Santiago et all. (Santiago et al., 2021) concluded that evidence about exercise therapy supports strength exercise for improve pain and function in the patient with LET. Specifically, wrist extension eccentric exercises and scapular stabilizers. This study also conclude that eccentric loading can be effective because the combination of muscle lengthening and contraction leads to increased tensile strength in the tendons.

Another systematic review and meta-analysis (Yoon et al., 2021) when comparing eccentric exercise with other strengthening exercises, eccentric exercise showed better effects in reducing pain in Achilles tendinopathy, likewise, eccentric exercise in hospital programs showed beneficial effects in pain reduction and functional improvement.

To summarize, the current evidence supports the use of eccentric exercise in LET. This does not contradict the fact that this type of exercise can be used in conjunction with others, such as those aimed at stabilizing the scapula.

References:

Bermingham, S. L., Sparrow, K., Mullis, R., Fox, M., Shearman, C., Bradbury, A., & Michaels, J. (2013). The Cost-effectiveness of Supervised Exercise for the Treatment of Intermittent Claudication. European Journal of Vascular and Endovascular Surgery, 46(6), 707-714. https://doi.org/10.1016/j.ejvs.2013.09.005

Bureau, N. J., Tétreault, P., Grondin, P., Freire, V., Desmeules, F., Cloutier, G., Julien, A.-S., & Choinière, M. (2022). Treatment of chronic lateral epicondylosis: A randomized trial comparing the efficacy of ultrasound-guided tendon dry needling and open-release surgery. European Radiology, 32(11), 7612-7622. https://doi.org/10.1007/s00330-022-08794-4

Bürge, E., Monnin, D., Berchtold, A., & Allet, L. (2016). Cost-Effectiveness of Physical Therapy Only and of Usual Care for Various Health Conditions: Systematic Review. Physical Therapy, 96(6), 774-786. https://doi.org/10.2522/ptj.20140333

Chen, Z., & Baker, N. A. (2021). Effectiveness of eccentric strengthening in the treatment of lateral elbow tendinopathy: A systematic review with meta-analysis. Journal of Hand Therapy, 34(1), 18-28. https://doi.org/10.1016/j.jht.2020.02.002

Page, P. (2010). A new exercise for tennis elbow that works! North American Journal of Sports Physical Therapy: NAJSPT, 5(3), 189-193.

Peterson, M., Butler, S., Eriksson, M., & Svärdsudd, K. (2014). A randomized controlled trial of eccentric vs. Concentric graded exercise in chronic tennis elbow (lateral elbow tendinopathy). Clinical Rehabilitation, 28(9), 862-872. https://doi.org/10.1177/0269215514527595

Purdam, C. R. (2004). A pilot study of the eccentric decline squat in the management of painful chronic patellar tendinopathy. British Journal of Sports Medicine, 38(4), 395-397. https://doi.org/10.1136/bjsm.2003.000053

Raman, J., MacDermid, J. C., & Grewal, R. (2012). Effectiveness of Different Methods of Resistance Exercises in Lateral Epicondylosis—A Systematic Review. Journal of Hand Therapy, 25(1), 5-26. https://doi.org/10.1016/j.jht.2011.09.001

Santiago, A. O., Rios-Russo, J. L., Baerga, L., & Micheo, W. (2021). Evidenced-Based Management of Tennis Elbow. Current Physical Medicine and Rehabilitation Reports, 9(4), 186-194. https://doi.org/10.1007/s40141-021-00322-7

Stasinopoulos, D., & Stasinopoulos, I. (2006). Comparison of effects of Cyriax physiotherapy, a supervised exercise programme and polarized polychromatic non-coherent light (Bioptron light) for the treatment of lateral epicondylitis. Clinical Rehabilitation, 20(1), 12-23. https://doi.org/10.1191/0269215506cr921oa

Williamson, A., & Hoggart, B. (2005). Pain: A review of three commonly used pain rating scales: Pain rating scales. Journal of Clinical Nursing, 14(7), 798-804. https://doi.org/10.1111/j.1365-2702.2005.01121.x

Yoon, S. Y., Kim, Y. W., Shin, I. S., Kang, S., Moon, H. I., & Lee, S. C. (2021). The Beneficial Effects of Eccentric Exercise in the Management of Lateral Elbow Tendinopathy: A Systematic Review and Meta-Analysis. Journal of Clinical Medicine, 10(17), 3968. https://doi.org/10.3390/jcm10173968

Round 2

Reviewer 2 Report

The authors have addressed all the queries raised by me. The manuscript can be accepted for publication.

Author Response

REVIEWER #2  RESPONSE (R2)

The authors have addressed all the queries raised by me. The manuscript can be accepted for publication.

Authors: Thank you so much. 
